# Wearable Skin Sensors and Their Challenges: A Review of Transdermal, Optical, and Mechanical Sensors

**DOI:** 10.3390/bios10060056

**Published:** 2020-05-28

**Authors:** Ammar Ahmad Tarar, Umair Mohammad, Soumya K. Srivastava

**Affiliations:** 1Department of Biological Engineering, University of Idaho, Moscow, ID 83844, USA; tara9039@vandals.uidaho.edu; 2Department of Electrical & Computer Engineering, University of Idaho, Moscow, ID 83844, USA; moha2139@vandals.uidaho.edu; 3Department of Chemical & Materials Engineering, University of Idaho, Moscow, ID 83844, USA

**Keywords:** wearable sensors, lab on a chip, MEMS

## Abstract

Wearable technology and mobile healthcare systems are both increasingly popular solutions to traditional healthcare due to their ease of implementation and cost-effectiveness for remote health monitoring. Recent advances in research, especially the miniaturization of sensors, have significantly contributed to commercializing the wearable technology. Most of the traditional commercially available sensors are either mechanical or optical, but nowadays transdermal microneedles are also being used for micro-sensing such as continuous glucose monitoring. However, there remain certain challenges that need to be addressed before the possibility of large-scale deployment. The biggest challenge faced by all these wearable sensors is our skin, which has an inherent property to resist and protect the body from the outside world. On the other hand, biosensing is not possible without overcoming this resistance. Consequently, understanding the skin structure and its response to different types of sensing is necessary to remove the scientific barriers that are hindering our ability to design more efficient and robust skin sensors. In this article, we review research reports related to three different biosensing modalities that are commonly used along with the challenges faced in their implementation for detection. We believe this review will be of significant use to researchers looking to solve existing problems within the ongoing research in wearable sensors.

## 1. Introduction

Advancements in miniaturization technology have resulted in a variety of wearable sensors, which are being used currently for various biomedical applications. Some of them have become a part of people’s everyday life. One example includes smart bands and smart watches integrated with heart rate monitors, pulse oximeters, accelerometers, gyroscopes, etc. [1,2,3]. Implantable devices have actually been used since the 1950s, when the first pacemaker was used to restore a normal heart rate by continuously pulsing only the ventricle [4]; then the evolution of implantable technology began leading to most recent implants such as Ocular and Cochlear implants [5,6,7]. Although biomedical implants do have great importance in advancing healthcare, they often require some expertise or a skilled technician to operate. In contrast, wearable technology is more user-friendly that may also be easily operable without the requisite medical or technical expertise [8]. The goal of wearable technology is to extract the information without involving any surgical procedures or implantation of materials that have a higher probability of leaving long-term side effects.

Recent advances in fabrication and packaging techniques enables to embed an enormous number of microelectronic devices and sensors on a chip at a very low cost. Moreover, with a blend of wireless 3G and 4G technology, these chips are widely used in different forms of physiological, biomechanical, biochemical sensing, monitoring and collecting biomedical data remotely [9]. This has significantly reduced the healthcare costs and has improved continuous monitoring of vital parameters, especially for athletes, thus improving their performance [10]. The future of remote biomedical sensing will be further transformed with the rolling out of 5G networks, combined with technologies such as the internet of things (IoT), machine learning, and artificial intelligence [11].

Today, physicians are using portable devices for diagnosis, such as glucometers, which provides instantaneous results and are generally invasive or minimal invasive in nature. However, completely noninvasive glucose monitoring technology is still in progress. Thus, the core idea here is to develop wearable sensors for these types of diagnoses that are non-invasive and hence, can be used in continuous monitoring systems. Furthermore, the population is increasing at an enormous rate, and there is shortage of physicians throughout the world [12]. This has forced the common man to seek out potential alternative solutions that would aid towards efficient usage of physicians’ time. Wearable technology can help individuals track their vital physiological parameters; with the option to consult a doctor only when needed [10,13]. A good indicator of the benefits of wearable technology is the day-by-day increasing demand for personal diagnosis and monitoring [14]. Its usage is currently not limited to monitor glucose levels, but has recently been proposed as an alternative method to perform fast HIV diagnoses [15], early detection of Alzheimer disease [16], and perspiration monitoring through wearable paper-based sweat sensor [17], thus leading personalized medicine to a new level [18].

Miniaturization of microelectronics along with stretchable microelectronics and telecommunication enables the production of wearable sensors leading to many novel clinical applications. Wearable technology consists of three processes: sensing, data processing, and wireless transmission of data [19]. Data processing is a crucial step because it involves noise reduction and feature extraction of clinically relevant physiological data [20]. One such example is the flexible electrocardiograph (ECG) sensor shown in Figure 1. Currently, sensing technology is not only limited to electrophysiological measurements such as ECG, electromyography (EMG), or electroencephalography (EEG), rather, it has expanded to electrochemical, electromechanical, and transdermal (minimal invasive by measuring fluid in between tissue) sensing [21]. 

This article aims to address the hurdles in extracting information from the body by reviewing some of the recently developed wearable sensors, their merits, construction, challenges, and the future outlook. There are many different types of wearable sensors available for various applications, but the discussion in this article is limited to only three major common types: transdermal, optical, and mechanical (piezoelectric, piezoresistive, piezocapacitive, and triboelectric). 

## 2. Skin: First Defense of the Immune System

The skin, largest organ of the human body, is made of soft tissue with an average weight of 4 kg and an area of 2 m^2^. The importance of skin is illustrated by the high mortality associated with extensive skin loss due to injuries such as burning [22]. Like most other organs, skin can also be divided into different layers: the hypodermis (deepest layer), the dermis, and the epidermis (superficial layer) as shown in Figure 2. The major part of the skin is the epidermis that is the uppermost or outer layer of skin. Underneath the epidermis lies the dermis, a vascularized region that provides support and nourishes the blood to distribute the cells in the epidermis. The dermis also has nerves, hair follicles, sweat glands, and sebaceous glands. The third layer and a deeper layer of the skin is the fat layer, also called subcutaneous fat [23]. 

The structure of the skin may differ slightly depending on the location, age of the person, illness, gender, etc. [22,24,25,26]. The epidermis (the outermost layer a skin) has a thickness ranging between 0.05 to 1.5 mm (thicker in palms and soles). It mainly consists of keratinocytes (keratin producing cells) that make up 95% of this layer and about 5% of other cells (Merkel, melanocytes, and Langerhans cells) [23,27], as shown in Figure 2. The epidermis can be divided from strata to dermis surface as: corneum, lucidum, granulosum, squamous, and basale. The basale stratum contains melanocytes, which produce melanin that gives color to the skin and protects the lower layers against the harmful effects of sun rays. More mature keratinocytes are present in squamous stratum and this layer, dendritic Langerhans, are used to alert the immune system, which is particularly attached to antigens in the damaged portion of skin. The lucidum and granulosum layers provide the main mechanical barrier because they have more mature keratinocytes that secrete proteins and lipids into the extracellular matrix, which results into a hydrophobic lipid envelope [23]. Going down from the surface, about 20 thin under layers, the dying keratinocytes are dry, merge without their nuclei and become stiff to form hexagonal-shaped corneocytes in the corneum stratum (the horny layer, so-called because they are like the horns of animals). Around the cells, in this layer, several sub-layers of fats are present (cholesterol, sphingolipids, fatty acids) to store ~3X the weight of the layer in water. Natural moisture present is about 30% and seems to burst when it drops to less than 10% because of the collapse of filaggrin into protein-specific enzymes that lead to control of osmotic pressure and water. The keratinocyte cycle, depending upon age, lasts from 28 to 50 days. The potential effect of the epidermis, in particular the stratum corneum, has been investigated in the general mechanical behavior of the skin [28,29,30].

### 2.1. Resistance to Mechanical Sensing

As discussed earlier, the human skin is the first line of defense; especially epidermis playing a pivot role in it. The special structure of the epidermis makes a barrier to collect information. It protects the underlying soft tissue from harmful ultraviolet radiation, and the corneum stratum is electrically resistive because of its oily nature [31]. Moreover, the epidermis is soft and stretchy that slides over the underlying soft tissues, producing a damping effect in response to mechanical forces making it more difficult to extract vital information (mechanical sensing). The complex anisotropic composition of the skin produces a nonlinear stress-strain curve in response to elongation. The Young’s modulus is highly variable and depends upon the hydration, age, and location [32]. It has been calculated that Young’s modulus of human skin is between 0.10 and 18.8 MPa at about 30% of strain within seven days of autopsy [33,34]. 

The human skin because of its frequency response to varying loads can be modeled as a spring-mass system, which consists of springs, masses, and dampers. At the point when the human skin is exposed to variable mechanical loads, the resistance of the skin to mechanical load changes in accordance with the frequency. As the recurrence of mechanical load increases, the resistance of the skin to that load (dampening effect) increases, and the skin’s elasticity (spring and mass effect) would become stiffer [35]. Moreover, the low and high frequency responses are distinct from each other. For example, at low frequencies, the shear waves spread along the surface of the skin. In contrast, shear waves proliferate through the mass medium in the dermis (that contains water-gel like segments called mucopolysaccharide) at higher frequencies. Shear wave engendering is transmitted using thick coupling inside the human skin medium [36]. Subsequently, water, which influences the viscoelastic properties of human skin (stratum corneum), can straightforwardly influence the mechanical properties of the inside physiologically with related frequencies [35,37,38].

### 2.2. Attachment of Sensors to the Skin

For the attachment of the sensors to the skin, first, an identification of the best match to its modulus is sought. An example would be silicone-based elastomer such as poly (dimethylsiloxane) (PDMS) that can be utilized as a sensor material. PDMS has a Young’s modulus of ∼3 MPa [39]. However, it is very firm in comparison with the human skin, which can prompt delamination. On the other hand, a milder or softer material, like Ecoflex, the silicone elastomer (Smooth-On) can be broadly utilized because of its Young’s modulus of 125 kPa that is closer to the human skin, taking into account better contact with the human body [40,41,42]. 

The common noises on wearable mechanical sensors can be grouped based on their sources: (a) noise due to movement, and (b) sensor inherent noise. The movement initiated noise is usually observed when applying mechanical sensors to a body when it is moving, for example, due to breathing or twisting impacts that produce significant pressure on the sensors [43,44]. These kinds of noises can be reduced by using a special type of sensors or modified algorithms that remove unwanted signal or noise when analyzing the extracted data [45,46]. Sensors have some own noises (inherent noise), adding to the challenges in wearable mechanical estimations. For example, the resistive sensors have temperature limitations whereas capacitive sensors have inherent parasitic noises that should be considered [47,48].

### 2.3. Resistance to Optical Measurements

Noninvasive biomedical measurements are usually obtained optically; the light source at a particular wavelength is exposed to the part of the skin where measurement is sought. The sensor measures the reflected and absorbed light or sometimes refracted light and thereafter the biomedical data can be characterized and quantified (the same phenomena as used by spectrophotometer). In transmission of an optical signal through the skin, the major factor is wavelength that determines, how deep the light can reach (Figure 3). The skin can likewise be used as a window to research the physiological state of the hidden organs. One such technique is the usage of functional near-infrared spectroscopy (fNIRS) [49] to observe the oxygenation changes in the human brain.

Sometimes, the skin offers a passive path for physiological data collection from hidden vascular structures and organs through its optical interfaces. Therefore it is important to evaluate, the skin’s optical properties when designing optical sensors [50]. Absorption, transmission, and scattering related to the human skin can be discussed by isolating the skin into three layers of particular optical characteristics [50,51]: (1) the stratum corneum, which is exceedingly keratinized due to the presence of dead squamous cells, (2) the hidden epidermis, which contains skin pigmentation (mostly melanin) that absorbs shorter wavelengths, for example UV, and visible light to some degree [52], and (3) the dermis, which is exceptionally vascularized and can be characterized using visible light, includes carotene, blood hemoglobin, and bilirubin [53]. Most of the visible light attenuates in the dermis because of its thickness as compared to the layers above as shown in Figure 3.

The optical attributes of the stratum corneum are basically characterized by its unpleasant surface, which results in diffuse reflection. Interfacial Fresnel reflection occurs 4–7% for typical light because of the difference in refractive index (n) of air (typically 1) to the stratum corneum i.e., ~1.55 [54]. In the epidermis, the light scatters with melanosomes (>300 nm in the distance across) [52], that shows mostly forward scattering, and within the melanin particles (30–300 nm in measurement) also called Mie scattering (the scattering of light with larger particles or about the same size of wavelength of light) [55]. In the dermis, scattering mainly results from collagen bundles and fibrils that create both Mie and Rayleigh scattering (the scattering of light with very small particles sometimes 10 times or more smaller than the wavelength of light) [55]. In short, the overall scattering of the skin is mainly due to the dermis layer because of its thickness that is considerably greater than that of the stratum corneum and the epidermis.

### 2.4. Stretching of the Skin

Another important challenge encountered in manufacturing strong mechanical sensors is material used in fabricating these sensors that extend and stretch along with the skin. The human skin has the inherent property of stretching to withstand large forces and aids in movement with flexibility. Any material, which is being used to attach to the surface of the skin must have the same or closely related properties of stretching, i.e., materials made of thin films that can withstand larger strains (ε = d/2r). Research has demonstrated that materials often fail due to high strain caused by slipping, breaking, or delamination of the thin film [56,57]. These happen because of the weak grip between the substrate and the film. Enhancing the grip of the substrate to the thin film with better adhesion essentially increases the mechanical properties due to strain delocalization [56,58,59]. Li et al. demonstrated in detail the significance of interfacial firmness between the substrate and thin film in strain delocalization theoretically [60,61]. Their results show that interfacial firmness enables the metallic film to deform consistently over huge strains, while weaker interfacial strengths prompt necking at zones of metal detachment and sometimes slipping from the substrate [61]. The adhesiveness can be increased without losing biosensing capability by the use of special polymers such as ethoxylated polyethyleneimine (PEIE) mixed with PDMS [62]. In brief, enhancing the bond of the sensor to the substrate will increase the reliability and dependability of the mechanical sensor.

## 3. Wearable Sensors

There are several types of wearable sensors currently in use. In this section, three major and commonly used types (i.e., transdermal microneedles, optical, and mechanical) will be discussed including their usage, construction, and recent developments.

### 3.1. Transdermal Microneedles 

Microneedles were at first created by Prausnitz and collaborators to upgrade penetration of medications and vaccines over the stratum corneum that anticipates the development of some charged, as well as polar operators over the skin [63]. Microneedles produce almost no pain or very little pain because they penetrate up to where no large ending nerves are present and are minimally invasive [64]. The prevalent utilization of microneedles to this point has been for transdermal drug and vaccine delivery [65]. But in recent times, researchers are focusing on expanding the application of microneedles as diagnostic sensors by accessing the interstitial fluid [66,67,68]. Transdermal sensing has great importance since they have the ability to obtain real-time clinical data for daily health monitoring. The schematic diagram of the porous microneedles system for fluid extraction is shown in Figure 4. 

#### 3.1.1. Extracellular Fluids in the Skin

The avascular region of the skin, the epidermis, can be normally separated into the stratum corneum and viable epidermis. The stratum corneum is viewed as a physical obstruction of dead cells with a “block and mortar” structure [70]. The viable epidermis has several epidermal layers, for which a specific enzyme and metabolic activity has been accounted for, with the exception of the recently reported biochemical activity inside the stratum corneum [70]. A critical difference between the viable epidermis and stratum corneum is the difference in polarity of these two layers; the viable epidermis is primarily hydrophilic, and stratum corneum is lipophilic in nature [71,72].

The dermis is the most vascularized area of the skin. Capillaries of dermis mostly feed blood to the dermis, with a little amount from venules and arterioles. Huge contrasts in healthy adults between the different constituents’ concentration in capillary blood have been reported [73]. An important fact to be noted here is that the blood contributes just 5% of the skin volume, and ISF (interstitial fluid) contributes to about 45% of the volume fraction with a higher proportion in the dermis as compared to subcutaneous tissue [69]. ISF is shaped by blood transcapillary filtration and excess fluid is taken away by the lymphatic vessels [74]. The exchange of fluid between the capillaries and the development of ISF is dictated by the capillary wall properties, hydrostatic pressure, and the concentration of protein in the interstitium and blood. ISF helps in the transportation of waste products and nutrients to the blood capillaries and cells, and antigens and cytokines to nearby lymph nodes contributing to the regulation of the immune system. Its organization and biophysical properties shift among organs and during tissue pathogenesis, remodeling, and inflammation [75]. The protein concentrations in ISF are about 33% lower than in plasma, with a lower concentration of magnesium, calcium, potassium, and sodium [76]. The most important element to consider here is the similarity of glucose in ISF and whole blood at steady-state conditions. Rapid changes in blood glucose levels in ISF at different lag times reflect glucose diffusion from the endothelium of capillary to ISF within the specific concentration gradient [77,78]. 

#### 3.1.2. Microneedle Insertion into the Skin

The productivity of the microneedles in recognizing analytes, either local or external to the inside of the skin, are subject to a viable and solid insertion of the needles into the skin tissue at a particular depth. To effectively accomplish this objective, a specific test is to be tailored to observe the insertion of the needles through the skin based on skin’s mechanical properties, basically Young’s modulus and puncturing stress. An interesting factor is that these mechanical properties have been found to change according to the skin strata, and furthermore, with the radius of the needle [79,80], and produced strain rate [81,82]. From a mechanical perspective, the skin is an anisotropic, heterogeneous, and viscoelastic material. Moreover, the dermis and epidermis of the human skin are diverse in nature and have different thickness as well as mechanical properties, which are still under investigation. Another important factor identified along with mechanical properties is its viscoelasticity, which has a greater significance in the insertion of microneedles into the skin [83]. To enhance the applicability of microneedles, hydrogel and polymer-based microneedles are used that allow rapid extraction of ISF. Mandal et al. recently developed a microneedle patch technology using biocompatible crosslinked coating on the surface of microneedle. When it is inserted into the skin, the coated microneedles would gradually swell to form a porous matrix for combined recovery of ISF and leukocytes [67]. Furthermore, the swellable polymeric needles were also developed by Chang el al. for timely metabolic analysis of ISF [66,84].

A noteworthy application of a microneedle-based sensor is the monitoring of blood glucose levels. Mukerjee et al. developed a variety of 200 empty silicon microneedles and showed interstitial fluid extraction from the human ear using wicking [85]. Glucose is then detected by a colorimetric glucose strip. The glucose level outcomes from the microneedle sensor were matched with the simultaneous detection of the subjects’ blood glucose, and the microneedle-based sensor results indicated similar level of accuracy. This demonstrates that a minimally invasive technique such as the microneedle can successfully extract ISF, which gives precise measurements of the glucose levels. Figure 5 shows the microneedle design and interstitial fluid extraction.

Different investigators have demonstrated the existence of a lag time associated with intravenous glucose level compared to ISF and it fluctuates between 6–45 min. For instance, F. Ribet et al. reported approximately 10 min of lag time between intravenous glucose and interstitial glucose for their continuous glucose monitoring system [68]. This lag time is subjected to different variables, i.e., blood flow rate, metabolic rate, sampling method, etc. [86,87]. Wang et al. analyzed this lag time and developed correlations between the interstitial glucose level and blood glucose level [88].

Recently a glucose detecting smart patch was developed that utilized, poly(3,4-ethylene dioxythiophene) PEDOT, a conductive polymer to entangle glucose oxidase specifically on the surfaces of strongly treated microneedle arrays of steel [89]. PEDOT provided an environment that is biocompatible and suitable to extract the dynamic enzyme and enabled glucose to diffuse into the polymer grid. Moreover, its electrical properties gave a low voltage transduction pathway. This novel technique allows the microneedles to utilize interstitial liquid without confounding complex microfluidic segments or sensor architectures [89].

Jina et al. also researched on microneedles that measured glucose continuously for up to 72-h in human skin [90]. The device comprised 200 empty silicon microneedles, each of which was 300 μm in length and 50 μm in lumen breadth, making an aggregate area of 6 × 6 mm^2^ [91]. This technique was applied to 10 diabetic subjects that precisely measured the glucose level with 15% MARD (mean absolute relative difference) in a 72-h period. In this ongoing research of microneedles, Li et al. developed a touch-activated microneedle-based sensor [92]. A single touch of a finger measured blood glucose through a nickel microneedle on a rabbit as a model.

### 3.2. Optical Sensors

Optical sensors are being utilized in a wide variety of applications [8,93,94,95,96]. Optical measurements can be taken from equipment such as the spectrophotometer, which are intended for use in a large-scale setup such as a clinic or a laboratory. On the other hand, measurements may be taken at a very small scale such as in electronic products including wrist smart watches (for heartbeat detection) and pulse-oximeter wearable sensors. In either setting, light is introduced into the body through the skin by the light source. Through the changes in the light properties (e.g., scattering or absorption), the body uncovers data by the light that is reflected on to the detector. The light sources range from single-wavelength laser to narrow bandwidth LEDs, depending upon the application [55]. Their wavelength can extend from deep infrared to UV, contingent upon the penetration depth requirement for the respective detecting application [97].

Optical sensor equipped gadgets are commonly available in the market nowadays. The most common optical sensors used in wearable electronics are used to detect oxygenation of blood and heart rate monitoring [98,99]. In diagnosing any clinical or biological event, the absorption and scattering properties of light concerning the location of the body, define the event. The most noticeable example is the oximetry technique that takes advantage of changes in the optical properties of hemoglobin in its deoxygenated and oxygenated state [99]. Analysis of the pulsatile part of bloodstream permits the estimation of key physiological parameters, for example, pulse inconstancy or variability through photoplethysmography (PPG) and oxygen saturation in arterial blood using pulse oximetry [100,101,102].

Photoplethysmography (PPG) is the optical technique that can be utilized to quantify blood volume changes in the microvessels of tissues (Challoner 1979). It has a broad clinical application and innovation, used in financially accessible medicinal gadgets, for instance, in pulse oximeters and digital blood pressure monitors for heart beat detection. The fundamental type of PPG only requires a couple of optoelectronic parts: a light source (typically mounted on the skin as shown in Figure 6A, to transmit light into the skin, and a photodetector to detect the light intensity variation related to the changes in volume (Figure 6B). PPG is usually a non-invasive technique and operates at a close infrared or red wavelength. The most perceived extracted features are the peripheral pulse waveform and is synchronized to every heartbeat (Figure 6C). The light interaction with biological tissue has already been investigated comprehensively by Anderson and Parrish in 1981 [51]. Some scientists have examined the optical procedures related to PPG estimations [103,104,105,106,107,108,109,110]. They have featured the key factors that can influence the measurement of light achieved by the photodetector: the movement of the walls of blood vessels, blood volume, and the red blood cell orientation. The example of PPG signal extracted from a different location in the body is given in Figure 6D.

One of the vital parameters in the optical detection is the wavelength of light source because of these three major reasons: (1) passage of light through water- the fundamental constituent, water, in tissues, have high light-absorbing characteristics in the longer wavelength regions of infrared and ultraviolet. Hence visible red region and near-infrared region can be used to penetrate more easily and allow measurement of blood flow and its volume. Thus, the red or closer infrared wavelengths are regularly dedicated in PPG light source [111]; (2) isosbestic wavelength- huge contrasts exist between oxyhemoglobin (HbO_2_) and hemoglobin (Hb) in absorption except at the isosbestic wavelengths [112]. For the measurement performed near 805 nm for the near-infrared region, the signal remains unaffected by blood oxygenation saturation change; and (3) penetration depth of the tissue- the penetration depth for a given sensing power threshold at a particular depth depends upon the working wavelength range [113].

Optical sensors, especially photoplethysmography, exhibits its extraordinary potential for use in an extensive variety of clinical measurements and detections. Currently, many researchers are exploring this technique due to its cost-effectiveness, portability, accessibility of its little semiconductor components, and simplicity or easiness to use in clinical settings. PPG-based innovation can be found in an extensive variety of economically accessible medicinal gadgets for measuring blood pressure, oxygen saturation, pulse, cardiovascular output, and its peripheral diseases. The success in this technology is still not absolute, challenges still exist, including its reliability, standardized measurements, and setting up regularizing ranges of data for correlation with patients, and for assessing different actions for treatment. Future research is additionally liable to see improvements in the measurements and detection, including home detection, PPG imaging, and basic endothelial evaluations [114,115,116].

### 3.3. Mechanical Sensors

Mechanical sensors are used to track the physical activity of the human body. Any irregular or even habitual movement can provide useful information about human health. The human gait is also considered as biometric property, tracking physical activity and periodic analysis of it can lead to detection of any abnormal patterns such as sudden tremors that are precursor of various diseases like Parkinson’s and Alzheimer’s disease. Hence, these mechanical sensors can contribute towards early detection of such fatal diseases [117]. Moreover, these sensors also have potential applications in rehabilitation, sports training, and prothesis [118]. The wearable mechanical sensors are widely used for detection of movements either on large scale (like arm, hand, and knee movement) or on small scale (like chest movement during breathing) [119,120]. 

In general, mechanical sensors can be categorized as stretchable strain sensors and pressure/tactile sensors. Stretchable strain sensors are used to detect motion through quantifying mechanical deformation by corresponding change in the electrical signal (piezoelectric or piezoresistive). The sensitivity or gauge factor (GFs) is an important parameter that influences the design of small-scale movement detection applications such as pulse monitoring and tracking respiration. However, the applications related to tracking the bending of joints in fingers, arms or legs demands high flexibility and stretchability of strain sensors. Some recent developments in strain sensors can be found in references [121,122,123,124,125,126,127,128]. Stretchable tactile sensors are designed to mimic the human perception of touch in response to external stimuli such as pressure, force, vibration, torsion etc. Human skin can sense as low as 5 Pa of pressure [129]. The human skin’s ability to sense this minute pressure and to withstand strong mechanical forces poses engineering challenges in designing flexible tactile sensors and fully functional stretchable electronic skin for health monitoring. Several stretchable tactile sensors have been developed recently [130,131,132,133,134,135]. The transduction principle that have potentially been used in these types of sensors are based on these effects: piezoelectric, piezocapacitive, piezoresistive, and triboelectric [131].

#### 3.3.1. Piezoelectric

Since last decade in material research, inorganic materials such as nanowire, nanoparticles, and nano-membranes are being considered for high-tech electronic devices on uncommon kinds of substrates, including elastic, rubber sheets, and plastic foils [136]. Further advance in wearable sensors can result in sensors that can easily fit in the human body and have the capacity to collect biological data [137]. One of its best known form is the piezoelectric-based mechanical sensor, which converts mechanical signal generated within the body to an electrical signal [138,139]. 

The mechanism of sensing of the piezoelectric sensor depends on the piezoelectric effect generated by the materials, producing electrical charges under mechanical stress, weight, or strain [138,140,141,142]. When a piezoelectric material comes under mechanical load or stress, the electrical polarization inside the material changes that leads to a change in surface charge or voltage at the piezoelectric material surface. Figure 7 better illustrates the working of piezoelectric sensors. They are mostly used as skin mounted tactile sensors i.e., for detection of bending of finger [138,140], body movement detection [143], detection of heartbeat by measuring arterial pulse, and other biomechanics characterization [144]. 

#### 3.3.2. Piezocapacitive

Piezocapacitive sensors are obtained by implanting conductive metal plates in adaptable materials, for example, poly(dimethylsiloxane) (PDMS) polymer. The conductive plates are commonly fabricated by utilizing deposition techniques like electroplating, evaporation, or sputtering [145,146,147]. Even though the polymeric packaging generally can sustain the chemical degradation and mechanical deformation, the conductive plates underneath with its interconnections can fail due to overstress and fatigue [148]. Indeed, even a little break in a plate or an interface can lead to the breakage of electrical connectivity, and hence failing the sensor [149]. 

The capacitive pressure sensor is generally applied to take advantage of its low-power consumption ability [150] and its long haul stability [151]. The structure of the capacitive sensor normally involves two parallel plates as an electrode with interlayer sandwiched between acting as a dielectric. Recently, silver nanowire (AgNW) integrated mesh emerged as a favorable electrode for capacitive sensors, because of its remarkable adaptability, cost-effectiveness, and simplicity of use [152,153,154]. In interlayer materials cases, the capacitance C depends mostly on the interlayer material dielectric permittivity ε_r_, the separation between the two plates, and the functional electrode area A:(1)$$C=\frac{\varepsilon_{0}\varepsilon_{r}A}{d}$$
where *ε*_0_ is the vacuum permittivity [155].

With the goal to enhance the sensitivity, extensive endeavors have been dedicated to the choice of appropriate interlayer material. Dielectric elastomers like polyurethane (PU) or PDMS have been utilized, since it is easy to change the distance ‘*d*’ by applying external pressure [151,156]. From Equation (1), aside from the parameters distance (*d*), and area (*A*), one can manipulate the dielectric permittivity (*ε_r_*) by using reasonable materials to change the sensitivity of the capacitive sensor. Utilizing the adaptable properties of graphene, a group of researchers have developed an extensive, stretchable, transparent touch capacitive sensor that would not just work in the multi-touch, scroll, and spread modes (as generally used in cell phones) but also recognizes the 3D shapes near its surface [157]. Figure 8A and B shows its layer by layer structure while 8C and 8D shows the change in capacitance for a multi-touch applied stimulus.

#### 3.3.3. Piezoresistive

Piezoresistive effect occurs when conductive materials exposed to mechanical deformation change their electrical properties due to the change in resistance. Because of the Poisson ratio (*v*), materials that are stretched likewise contract in the transverse direction of extension. Thus, the resistance R for a conductive material is given by:(2)$$R=\frac{\rho L}{A}\;$$
where, L is the length, ρ is the resistivity, and A is the cross-sectional area of the conductor. The piezoresistive effect is generally used for the detection of the physiological movement of humans in the form of wearable electronics because of its simple design, high sensitivity, and simple readout [158,159].

A functioning resistive sensor in wearable technology must have adaptability, stretchability, sensitivity, and low hysteresis [160,161]. A device that can stretch and is flexible will be mechanically robust during attachment on to the body and can be used for longer terms. In wearable electronics, the strain sensor usually should not exhibit broad plastic deformation during repeated or continuous strain. In particular, strain sensors must have the sensitivity to strain to enhance signal extraction and dynamic strain detection. The sensitivity of strain is usually described by a measuring factor (GF): (3)GF=ΔRR0ΔLL=ΔRR0ε
where ΔR is the adjustment in opposition, $R_{0}$ is the unstrained obstruction, and ε is the strain for change in length ΔL as compared to its original length L. In order to measure or monitor human movement, the sensor should be flexible and stretchable within the capacity of maintaining high sensitivity at large dynamic range [162,163]. Typically strain sensors are developed using patterned thin-film on flexible elastomer substrates. With the advancing research in materials, now strain sensors are fabricated using nanowires, nanotubes, nanoparticles, and graphene. However recently, carbon nanotubes (CNTs) were used to make a highly flexible sensor (Figure 9), which shows promising results under high strain rates [41].

Figure 9 illustrates the superior performance of a CNT-based sensor placed at separate joints. Figure 9a shows a thin CNT film on the substrate. Figure 9b–d illustrate the placement of the sensor on the finger, elbow, and knee, respectively showing the corresponding changes in resistance values. Initially all sensors in the original position are set to straight with respect to each joint. The joints are then bent, and then returned to the original position to evaluate the performance under strain. The success of the resistance value returning to the original baseline value in all the three scenarios demonstrates the potential effectiveness of the sensor for epidermal use. To study more of the recently developed strain sensors, the readers are referred to references [48,164,165,166,167,168]. 

#### 3.3.4. Triboelectric 

With the development of wearable smart electronics, there is immense need of self-powered systems for the purposes of durability, portability, and comfort. Researchers, with the help of nanotechnology, have developed various nanogenerators [169] including the triboelectric nanogenerator (TENG) invented by Prof. Wang’s group in 2012 [170]. Triboelectric effect is the electrification of a material induced by friction; it becomes electrically charged after coming into frictional contact with a different type of material. TENG is based on the coupling effect of electrostatic induction and triboelectrification. The working principle can be explained by simple composition of TENG with two materials having electrodes fabricated on one side separated by elastic spacers as shown in Figure 10. In response to the external mechanical force or deformation, triboelectric charges are produced on the surfaces due to contact. Once the mechanical force is released, the surfaces gets separated producing a gap between them, and the opposite charges on the surfaces produce a potential difference [171,172]. Therefore, it can also be considered as a mechanical energy harvesting device because it has the capability to monitor touch stimuli without any external power source [173].

The output of triboelectric effect depends upon the relative position of the two materials that contribute different polarities for electrostatic induction. Moreover, the surface structure of the materials also plays an important role. An increase in roughness produces more friction resulting in an enhanced electrical output [174,175,176]. Recently, various TENGs have been invented based on different triboelectric materials, substrates, spacers, and electrodes [173,177]. However, polymer films are considered as the best choice because of their flexibility, light weight, robustness, and better triboelectric output [177,178,179].

## 4. Conclusions

In this review, three important and most commonly utilized wearable skin sensors, i.e., transdermal needles, optical, and mechanical are discussed, specifically related to the challenges encountered during biosensing. Although the progress in wearable technology has been significantly accelerated by advancement of research in materials sciences and printed electronics, challenges still remain. The biggest hurdle that these sensors are facing is the inherent protective structure of the skin that creates a barrier between the sensors and the available information below the surface. Some of the skin sensors, like stretchable mechanical sensors, also need additional disposable components such as adhesives that are functional as long as the stratum corneum refreshes itself completely. In many cases, it is difficult to last longer because of bathing, skin oils, irritation, etc. Therefore, to produce these sensors commercially, these challenges must be considered, and we need to come up with procedures and techniques that make attaching and detaching easy. 

Self-calibration of wearable devices is another challenge since every person is unique and various other factors (diet, family history, etc.) responsible for personal health needs to be considered. Hence, the early symptoms for disease diagnostics might change for different people based on the above factors. Thus, the self-calibration of wearable devices with incorporation of machine learning technology for analysis of data can provide accurate health monitoring. Another technical challenge is the robustness and durability at varying weather-related conditions (humid, wet, warm, etc.) and during different activities (swimming, running, sunbathing, etc.). This can be addressed by a new smart design and suitable material to build sensors that can tolerate such conditions. 

In short, it is important that researchers overcome the skin-penetration problem while developing sensors and incorporate some form of communication technology. We expect that this review has given researchers a guide on how to fill those gaps by pinpointing the challenges for the specific type of sensor they are developing pertaining to the application.

## Figures and Tables

**Figure 1 biosensors-10-00056-f001:**
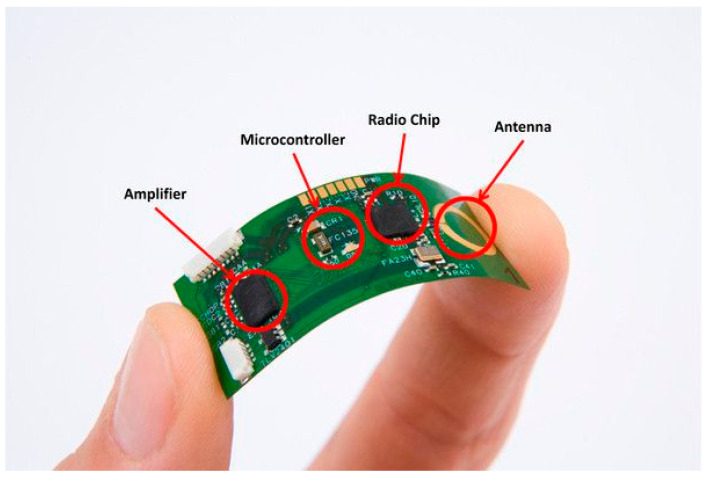
Flexible electrocardiograph (ECG) sensor with its components (courtesy of IMEC, the Netherlands).

**Figure 2 biosensors-10-00056-f002:**
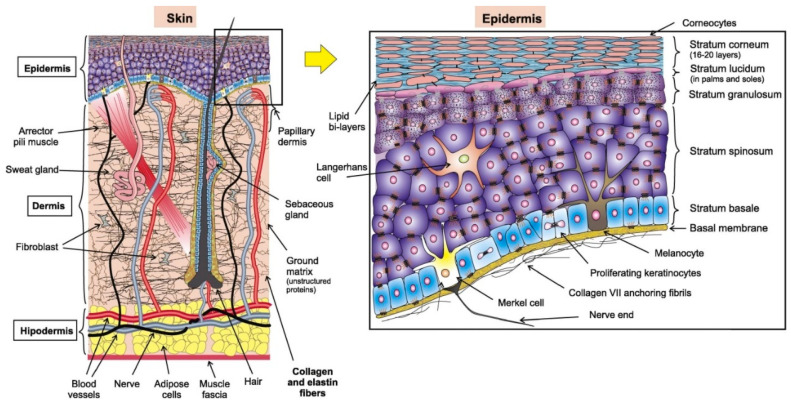
Illustration of the different layers of the skin.

**Figure 3 biosensors-10-00056-f003:**
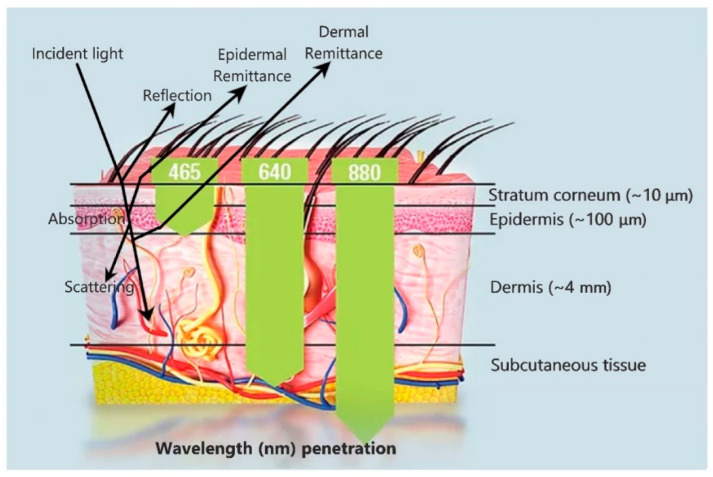
Incident light shows reflection, absorption, and scattering phenomena at different locations in skin layers, (Green downward pointing arrows). Light penetration into the skin with respect to its wavelength.

**Figure 4 biosensors-10-00056-f004:**
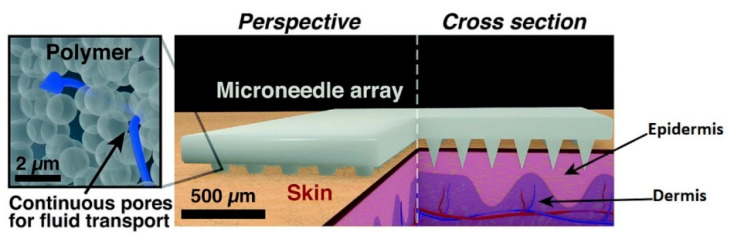
Schematic diagram of microneedles inserted into the skin for interstitial fluid extraction [69].

**Figure 5 biosensors-10-00056-f005:**
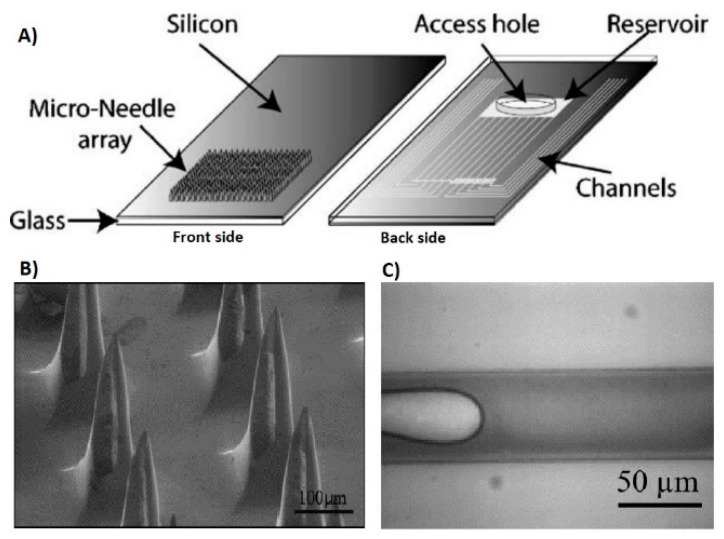
(**A**) Structure of microneedle Array, (**B**) SEM image of Snake Fang microneedle design, (**C**) Transdermal extraction leading to interstitial fluid in micro channels [85].

**Figure 6 biosensors-10-00056-f006:**
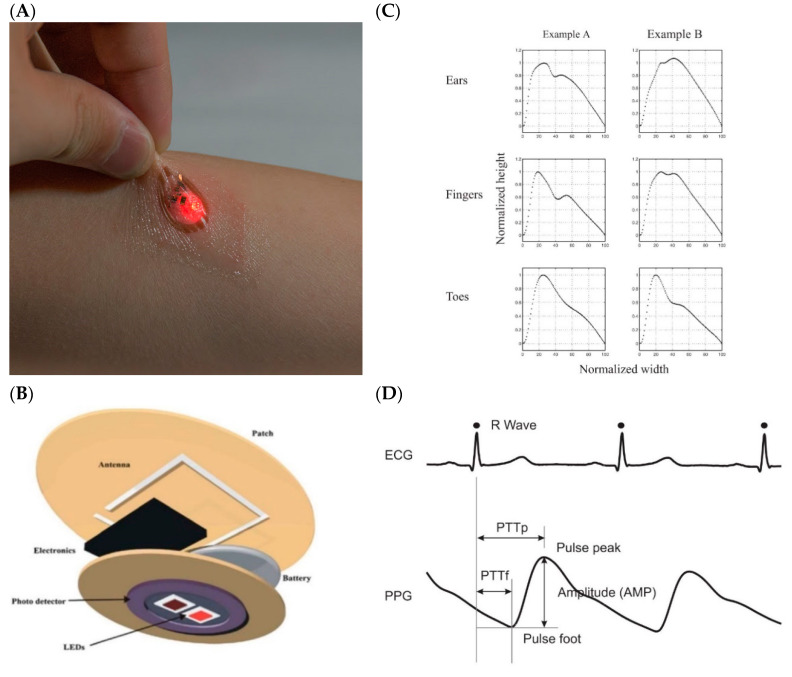
(**A**) Optical sensor mounted on skin, (**B**) Basic diagram of a wireless photoplethysmography (PPG) sensor, (**C**) PPG signal collected from three different locations, (**D**) Comparison of the PPG signal with ECG signal.

**Figure 7 biosensors-10-00056-f007:**
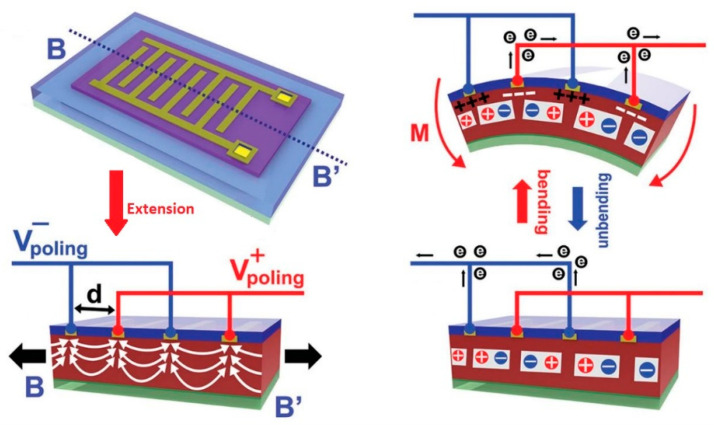
Schematic Diagram of working of flexible Piezoelectric sensors [140].

**Figure 8 biosensors-10-00056-f008:**
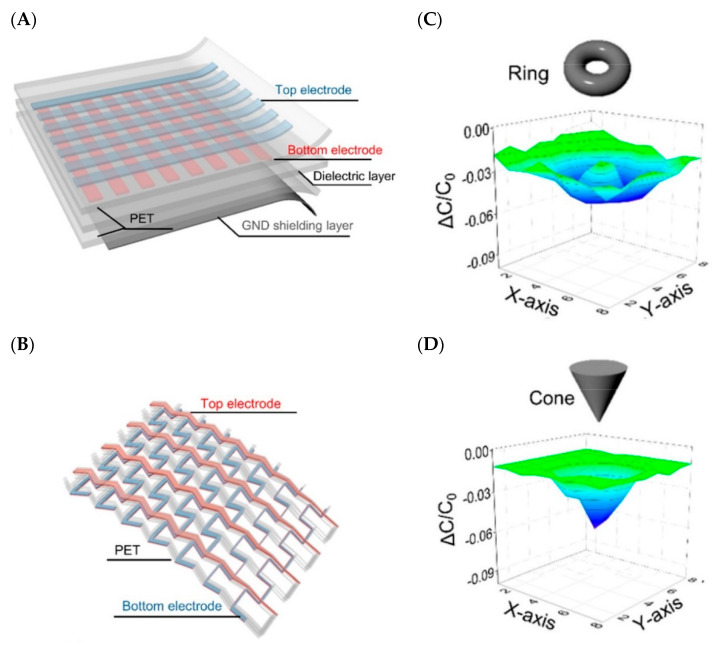
(**A**) Graphene-based flexible capacitive sensor, (**B**) The mesh-like internal structure shows the top electrode as red and bottom as blue, (**C**) Detection of Ring shape, (**D**) Detection of Cone shape [157].

**Figure 9 biosensors-10-00056-f009:**
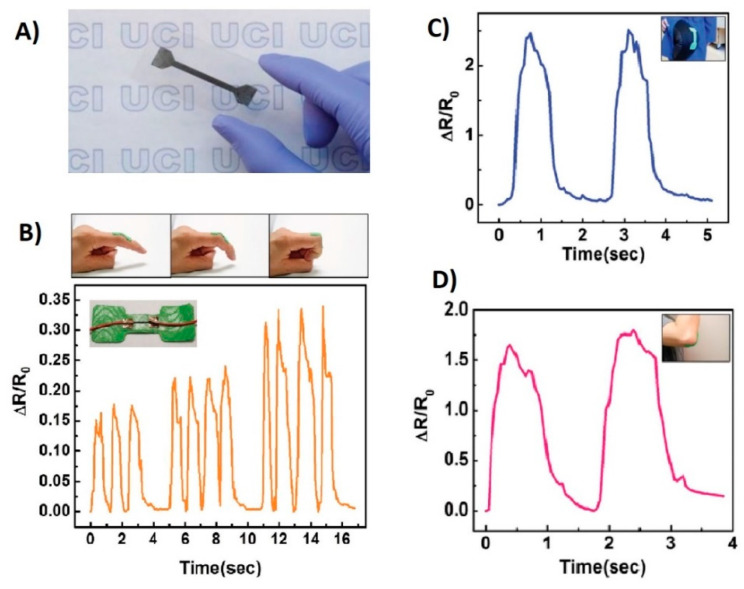
(**A**) Flexible carbon nanotubes (CNT)-based piezoresistive strain sensor, (**B**) change in relative resistance due to bending of the finger, (**C**) due to the movement of knee joint, (**D**) elbow joint [41].

**Figure 10 biosensors-10-00056-f010:**
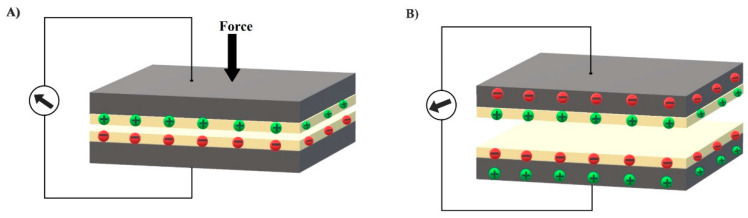
Flexible triboelectric sensor’s schematic illustration (**A**) transduction principle when force is applied, (**B**) potential difference produced when force is released.

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
