# Peer review of "Wearable Skin Sensors and Their Challenges: A Review of Transdermal, Optical, and Mechanical Sensors"

_biosensors, 2020, doi:10.3390/bios10060056_

Round 1

Reviewer 1 Report

This manuscript reviewed several wearable sensors for biomedical applications, which first introduced the properties of human skin including resistance to mechanical and optical sensing, and stretchability. Afterward, three main types of sensors including transdermal microneedles, optical, and mechanical sensors had been fully discussed. After concluding the recent progress, the authors also indicated the current challenge and perspective in the future. This topic is interesting, and this manuscript is well written and organized. However, there are several issues need to be solved before publication.

(1) Most of the cited literature is too old to show the recent progress in the wearable sensors field. It seems that this manuscript was prepared five years ago, and just a few new citations were added.

(2) In line 77, the “m2” should be “m2”.

(3) The caption of Figure 2 is the same as that of Figure 1.

(4) The format of citation 29 (line 113), 30 (line 117), 37 (line 134), 51 (line 165), 71 (line 238), 102 (line 325), 140 (line 423) should be corrected.

(5) The caption of Figure 5 is the same as that of Figure 4.

Author Response

Summary of Modifications

  • The authors have reviewed the writing of the paper and fixed the grammatical errors and typo errors of citations.
  • Technical Question asked by the reviewers have been addressed.
  • New material has been added with most recent development in the field and updated the citations accordingly

Important Note:

The reviewers’ original comments are in black regular font. The authors’ comments are in blue regular font and corresponding changes in the manuscript are in regular red font. Explanations about manuscript additions if needed, are provided just prior to the changes, and are given in black bold italic font. The all changes in the manuscript is highlighted to track easily.

Reviewer 2 Report

It is required to rephrase for a better understanding the following sentences:

  • L18: On the other hand, biosensing is not possible overcoming this resistance
  • L120-121: The human skin can also be demonstrated as dampers, springs, and masses because of its frequency dependency
  • L124-127: Moreover, the shear waves spread along the surface of the skin at low frequencies, while shear waves proliferate through the mass medium in the dermis (that contains a mucopolysaccharide– water gel segments) at higher frequencies
  • L157-159 “Sometimes, the skin offers a passive path to study the optical interface of hidden vascular structures and organs; therefore, the skin’s optical properties are important to study when designing optical sensors.”
  • L188-189 “Research has demonstrated that strain fails in materials occur because of slipping, breaking, or delamination of the thin film”
  • L286:” Optical measurements are being taken on a wide scale”

 Further comments:

  • L36-37: Layman? There is a lack of proper referral towards general populations.
  • L 39: From the sentence stated, it looks like any surgical procedures or implantation of materials result in always in side effects.
  • L47: There are many published papers on non-invasive glucometers and many commercial available devices are already on the market (i.e. The Progress of Glucose Monitoring-A Review of Invasive to Minimally and Non-Invasive Techniques, Devices and Sensors, Sensors (2019).
  • L55: HIV-1 sampling is still based on blood sampling even in the paper referred by authors.
  • L63: the acronyms of ECG appears for the first time here and it is not explained.
  • L77-78: Redundant! Do not be repetitive with the concepts.
  • L132: if it is attached...how can it be INTO?
  • L211: Can the author provide any example or reference? “But in recent times, researchers are focusing on expanding the application of microneedles by accessing interstitial fluid that allows being used as diagnostic sensors”
  • L255-262: glucose readings from a sensor sampling in interstitial fluid differ substantially from blood glucose. In the paper cited by the authors, it is shown that the ISF and blood sample gave the same colorimetric change from clear to deep blue indicating an approximate glucose concentration of 80–120 mg/dl glucose. Which is not really what the authors speculate as “indicated greater accuracy”
  • L322: what does the author mean with “longer infrared”?
  • Where Figure 3 has been taken from? Figure 8 is not described in the text.
  • References 29; 34; (Sylgard 184, 10:1)37; 51; 71; are not cited properly in the text.

This reviewer rejects the paper because there is a lack of research on bibliography and the writing should be absolutely reviewed by an English corrector.

Author Response

(The authors gave the same response as above.)

Reviewer 3 Report

The review paper introduces the general fundamentals of human skin and the progress of several types of skin sensors. It is of significance from an aspect of human skin structure and property considerations for the development of wearable sensors, which is usually ignored in material scientists in making sensors.  The manuscript is developed succinct and clear.  Thus, it is suggested to be accepted for publication with the following suggestions:

(1) The title "Wearable skin sensors and its challenges" is broad since it doesn't cover all kinds of wearable skin sensors, there are still a lot of others to be noted such as temperature sensor, biosensor; so please make the title more specific and a little narrow.

(2) For the mechanical sensors, it can be divided into stretchable strain sensors and pressure/tactile sensors; in addition, another transduction mechanism of the triboelectric effect is developed recently, it is suggested to include this type of sensors in the review.

(3) Please check carefully the format and typo of the manuscript, there are some mistakes, especially the cited references.

Author Response

(The authors gave the same response as above.)
